# Progress in Research and Prospects for Application of Precision Gene-Editing Technology Based on CRISPR–Cas9 in the Genetic Improvement of Sheep and Goats

Zeyu Lu [1,2,3], Lingtian Zhang [4], Qing Mu [1,2,3], Junyang Liu [1,2,3], Yu Chen [1,2,3], Haoyuan Wang [1,2,3], Yanjun Zhang [1,2,3], Rui Su [1,2,3], Ruijun Wang [1,2,3], Zhiying Wang [1,2,3], Qi Lv [1,2,3], Zhihong Liu [1,2,3], Jiasen Liu [5], Yunhua Li [5] and Yanhong Zhao [1,2,3,*]

1 Key Laboratory of Mutton Sheep Genetics and Breeding, Ministry of Agriculture and Rural Affairs, Hohhot 010018, China; 15248157808@163.com (Z.L.); 18353621226@163.com (Q.M.); 15771343186@163.com (J.L.); 19847372538@163.com (Y.C.); 17795992714@163.com (H.W.); imauzyj@163.com (Y.Z.); suruiyu@126.com (R.S.); nmgwrj@126.com (R.W.); wzhy0321@126.com (Z.W.); lvqi1202@imau.edu.cn (Q.L.); liuzh7799@163.com (Z.L.)
2 Inner Mongolia Key Laboratory of Animal Genetics, Breeding and Reproduction, Hohhot 010018, China
3 College of Animal Science, Inner Mongolia Agricultural University, Hohhot 010018, China
4 Cofco Jia Jia Kang Food Co., Ltd., Songyuan City 131500, China; zhanglingtiancofco@163.com
5 Institute of Animal Husbandry, Inner Mongolia Academy of Agricultural and Animal Husbandry Sciences, Hohhot 010031, China; jsliu588@163.com (J.L.); yhli5277@163.com (Y.L.)
* Correspondence: 13947196432@163.com; Tel.: +86-13947196432

**Abstract:** Due to recent innovations in gene editing technology, great progress has been made in livestock breeding, with researchers rearing gene-edited pigs, cattle, sheep, and other livestock. Gene-editing technology involves knocking in, knocking out, deleting, inhibiting, activating, or replacing specific bases of DNA or RNA sequences at the genome level for accurate modification, and such processes can edit genes at a fixed point without needing DNA templates. In recent years, although clustered regularly interspaced short palindromic repeats (CRISPR)/Cas9 system-mediated gene-editing technology has been widely used in research into the genetic breeding of animals, the system's efficiency at inserting foreign genes is not high enough, and there are certain off-target effects; thus, it is not appropriate for use in the genome editing of large livestock such as cashmere goats. In this study, the development status, associated challenges, application prospects, and future prospects of CRISPR/Cas9-mediated precision gene-editing technology for use in livestock breeding were reviewed to provide a theoretical reference for livestock gene function analysis, genetic improvement, and livestock breeding that account for characteristics of local economies.

**Keywords:** precision gene editing; CRISPR–Cas9; livestock genetic breeding; cashmere goat; editing efficiency; fixed point integration

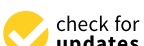



## 1. Introduction

In the 1990s, genome-editing technology experienced significant development, and thanks to the rapid innovation and refinement of this technology, gene-editing methods that accurately targeted modification genomes emerged [1,2], such as transcription activator-like effector nuclease (TALEN) and zinc-finger endonuclease (ZFN), while clustered regularly interspaced short palindromic repeats (CRISPR) appeared after 2000 [3]. Due to the advantages of CRISPR, such as its low cost, simple operation, accurate editing, high efficiency, fast development rate, stable structure of the Cas9 nuclease, easy design, system maturity, and ability to simultaneously target multiple gene sites [4–6], CRISPR is the most effective of all such technologies. It is widely used in livestock and poultry gene function analysis, molecular breeding, genetic improvement, disease resistance breeding, molecular biology and molecular cytology, and other genome editing fields [7–9]. CRISPR–Cas9, a genetic

tool, is a vital 21st-century innovation [10–13]. Table 1 summarizes the advantages and disadvantages of the three gene-editing techniques [13,14]. In addition, epigenetic traits that affect livestock genetics and breeding are regulated by genes. To breed better livestock varieties and improve animal husbandry's productivity, it is necessary to accurately edit the genome with specificity and by using multiple loci simultaneously. In this study, CRISPR–Cas9 technology and its applications in livestock and poultry breeding were studied to provide a reference for the use of CRISPR–Cas9 technology to study new breeding methods, such as genome transcriptional regulation and epigenetics in large livestock.

**Table 1.** Comparison between advantages and disadvantages of CRISPR–Cas9, TALEN, and ZFN gene-editing technologies.

| Characteristic | CRISPR–Cas9 | TALEN | ZFN |
|---|---|---|---|
| Price | Low | High | Low |
| Precision | Pinpoint | Moderate | Low |
| Combination mode | RNA–DNA | Protein–DNA | Protein–DNA |
| Design and construction | Easy | Difficulty | Moderate |
| Target fragment size | 20–50 bp | 30–40 bp | 18–36 bp |
| Application | Wide | Small | Small |
| Off-target effect | High | Low | Low |

## 2. CRISPR–Cas9 Gene-Editing Technology

### 2.1. Overview and Principles of CRISPR–Cas9

CRISPR is an adaptive immune system present in bacteria and archaea. It is composed of the Cas9 protein, functional genes, short regularly clustered repeat sequences, and similarly long spacer DNA sequences arranged from 5′ to 3′ [15,16], effectively defending against the invasion of exogenous viral DNA [17] and protecting genetic information from destruction [18]. The proteins encoded using CRISPR are called Cas (CRISPR-associated proteins), which is a nucleic acid endonuclease composed of a nuclease domain (RuvC-like) and nuclease functional regions (McrA-like HNH), and it mainly encodes functional proteins binding to nucleic acids [19]. According to Cas proteins' different mechanisms of action, the CRISPR–Cas9 system is divided into two classes and six types [20]. Class 1 includes types I, III, and IV, and class 2 includes types II, V, and VI. Among them, the CRISPR–Cas9 system of type II is the most advanced gene-editing technology used to improve the genetic breeding, reproductive performance, and nutrient intake levels of livestock due to its simple structure, strongest specificity, and highest efficiency [21]. The type II CRISPR–Cas9 system's mechanism of action is based on the principle of base complementary pairing, where the Cas9 protein is guided by a single-guide RNA (sgRNA) to recognize the PAM sequence and targeted cleavage is performed on the target DNA to cause double-strand break (DSB), stimulating the repair mechanism of DNA to repair the DSB site through the non-homologous end junction (NHEJ) or homologous directed repair (HDR) [22].

The Cas9 protein undergoes cleavage through the combined action of specific CRISPR RNAs (crRNAs) and trans-activating CRISPR RNAs (tracrRNAs) via base-pair binding [23]. Replacing crRNA and tracrRNA with sgRNA [24] simplifies the CRISPR–Cas9 system's operation. sgRNAs are adjacent to the protospacer adjacent motif (PAM) [25], ensuring that the Cas9 protein accurately recognizes exogenous DNA without destroying its own genetic material. However, not all sequences are efficient and specific for induction cutting. Currently, different PAM (5′-NGG-3′, N as A/T/C/G) forms of CRISPR–Cas9 systems are used depending on the editing sites' sequence characteristics [26]. The CRISPR/Cas system immunoregulates invading elements in three stages: acquisition, transcription, and interference [27]. The system is also divided into two different subsystems: the highly conserved "information processing" subsystem (acquisition) and the "implementation" subsystem (transcription and interference). The first stage involves obtaining recognition and inserting the ingested PAM into the dominant or adjacent area [6]. In the second stage,

the CRISPR sequence is transcribed together with PAM to form a long transcript called pre-CRISPR RNA, and mature small crRNAs are processed using a unique Cas protein and combined with tracrRNAs to form complementary chains [28]. In the third stage, the crRNA-tracrRNA complex accurately induces the Cas9 protein-targeted cleavage of the exogenous DNA.

CRISPR–Cas9 generally repairs genetic material via non-homologous end joining (NHEJ) and homologous recombination (HR). NHEJ repair often leads to the random insertion or deletion of small fragments, as well as chromosomal rearrangement. HDR repair is achieved by artificially synthesizing sgRNAs instead of the crRNA–tracrRNA complex so that Cas9 proteins can function at the target site [29]. This operation treats Cas9 and sgRNA as two separate systems subject to insertion, deletion, or replacement at their respective targets. It also changes the Cas9 cleavage site by modifying sgRNAs, making CRISPR–Cas9 technology more widely applicable. In addition, Wang et al. [30] developed a genome-editing strategy using Cas9/sgRNA-mediated allele exchange to treat compound heterozygous mutant diseases detected in recessive inherited diseases. This method is flexible in design and compatible with different mutations, and it is safer and does not require external DNA repair templates to prevent unwanted changes. NHEJ is mainly used to study gene knockout processes such as the insertion, deletion, or chromosomal rearrangement of small fragments, making it the first method used with the CRISPR–Cas system that can prevent the functioning of endogenous genes in domestic animals such as goats [31], sheep [32], cattle [33], and pigs [34]. When using the homologous template of heterologous mutated DNA as the repair template of DSB, HDR can insert exogenous DNA fragments into specific locations of the genome to accurately edit the target sites [35]. Following NHEJ, HDR achieved gene knockout in large animals for the first time. Although HDR can be accurately repaired, it has certain drawbacks. Randomly introduced insertions or deletions may cause gene knockout, new functions may be generated, and even activating DNA damage regulatory mechanisms may induce cell apoptosis [36].

Following recent research into the powerful Cas9 protein [37], ever-more CRISPR–Cas9 tools [38] have emerged, such as Streptococcus pyogenes Cs9 (SpCas9), the main tool used for studying animal gene editing [39]; the "BE", which causes the C and T to replace each other [40,41]; and the "PrimeEditor" tool [42], which can accurately perform targeted editing without using the template of the donor genetic material. Not only do these new tools have a low probability of missing their target, but no new functional changes will occur [43].

### 2.2. Comparison between CRISPR–Cas9 and Other Gene-Editing Techniques

At present, the mainstream gene-editing technology is the CRISPR–Cas9 system, mediated by a shorter RNA. In the past, research into gene-editing techniques using zinc-finger nucleases (ZFN) and transcript activator-like effector nucleases (TALEN) was more common. ZFN and TALEN both use the Fok I family-restrictive endonuclease that forms dimers to specifically recognize and non-specifically cut the targeted DNA [44]. However, ZFN recognizes three nucleotide sequences, while TALEN only recognizes mononucleotides with variable double residues formed via multiple tandem repeat sequences. Due to this feature, TALEN is not affected by external factors during the design stage, and TALEN can target longer sequences [45], meaning that it has the advantages of a short preparation cycle and low experimental costs, making it more favorable than ZFN. Notably, the length and flexibility of the DNA binding domains in TALENs are more easily customized than those of ZFNs. Like CRISPR–Cas9, they repair genetic material through the HR and NHEJ mechanisms. Compared with traditional gene-editing techniques, ZFN and TALEN regulate gene expression through targeted insertion, base deletion and replacement, transcriptional inhibition or activation, etc., and improve gene targeting efficiency [46]. However, the development of these technologies has been limited thus far due to their complex operation and high cost.

However, compared with the above two technologies, CRISPR–Cas9 has incomparable advantages for livestock genetics and breeding. Table 1 summarizes the key differences between the three genome-editing technologies. Firstly, ZFN is composed of zinc-finger protein (ZFP) and the I1S-type restriction endonuclease Fok I, which form a specific domain for binding to DNA targets. TALEN specifically binds to DNA targets through transcription-activating effector-like factors (TALE), both of which induce Fok I protein dimers to cleave DNA targets. Constructing the CRISPR–Cas9 system is extremely simple, requiring no significant human or material resources to design, process, and screen proteins that recognize DNA. The design and synthesis of a gRNA that complements DNA are required to guide the Cas9 protein to perform DNA double-stranded cleavage [47], which is why this technology is widely used across this scientific research field. Secondly, the CRISPR–Cas9 system simultaneously edits multiple independent target gene sites based on multiple gRNAs at different sites without needing multiple domain connections [48]; thus, it is highly efficient. ZFN reduces mismatch rates and off-target effects by creating a cutting enzyme to produce DNA DSB from nucleases, which is then repaired via HDR; purified zinc-finger nuclease protein can also be directly introduced into cells to reduce off-target effects. TALEN can also replace nucleases with other functional enzymes to reduce the cytotoxicity stemming from off-target effects. Although CRISPR–Cas9 is highly efficient and specific, due to its large genome, sgRNA can also produce non-specific mutations that match non-target sites, causing off-target effects. Therefore, past researchers calculated and designed suitable editing sites to improve the specificity of sgRNA [46], controlled the number of sgRNAs [49], and continuously created new Cas9 protein mutants, such as EvoCas9 [50], Sniper Cas9 [51], xCas9 [52], and AsCpf [53]. These gene-editing tools, due to their simpler structures and higher editing efficiency [44], can be flexibly used in any nucleic acid sequence. Further innovation and refinement will ensure continuous improvement of CRISPR–Cas9's editing ability.

Overall, although other gene-editing technologies have some advantages, the CRISPR–Cas9 system has revolutionized its fields of application due to its simple design and operation, low cost, ability to edit multiple genes simultaneously, and high mutation efficiency [8,54].

## 3. Application of the CRISPR–Cas9 System in Livestock and Poultry

### 3.1. Application of CRISPR–Cas9 Technology in Pig Genetic Breeding

As pigs are an important livestock for meat production, pig farming is a gradually expanding industry, and intensive breeding has seen different novel diseases emerge. It is vital that researchers cultivate genetically modified pigs with disease resistance. Pigs were the first large farmed animals edited via CRISPR–Cas9 [55], and CRISPR–Cas9 gene-editing technology provides new ideas for enabling genetic improvement and instilling disease resistance via breeding. Xiang et al. [56] used CRISPR–Cas9 to knock the non-coding region of insulin-like growth factor 2 (IGF2) into Bama pigs. Through slaughter experiments and the calculation of daily weight gain and feed-to-weight ratios, they found that without changing the health of Bama pigs, meat production can be improved, with excellent meat quality as a result. Mitochondrial uncoupling protein 1 (UCP1) is an important molecular marker of brown adipose tissue in animals. The non-chestnut thermogenesis and cold resistance of brown adipose tissue actually rely on UCP1 protein-mediated uncoupling respiratory thermogenesis. Therefore, UCP1 can regulate and maintain the body temperature of piglets in cold environments, reducing newborn piglets' body fat production rate as well as improving their lean meat and birth rates [57]. The process of using CRISPR–Cas9 to knock UCP1 into their chromosomes to breed healthy lean pigs with cold resistance and low fat will reduce economic losses in both agriculture and animal husbandry [58].

Classical swine fever virus (CSFV) and porcine reproductive and respiratory syndrome virus (PRRSV) are both envelope single-stranded sense RNA viruses [59], and their replication does not produce intermediate products. Transgenic, CSFV-resistant pigs were bred using CRISPR–Cas9, and researchers found that the growing pigs, pregnant sows,

and their fetuses were all resistant to PRRSV infection [60]. CD163 knockout pigs were also bred, and the researchers found that the replication of CSFV was restricted while disease resistance was stably transmitted to offspring. The African swine fever virus (ASFV) has a higher mortality rate than ordinary swine fever, reaching 100% in some cases. Hubner et al. utilized CRISPR–Cas9 to target ASFV phosphoprotein p30 to reduce the number of African swine fever viruses [61]. Xu et al. [62] constructed CD163 and porcine aminopeptidase (pAPN) double-gene knockout (DKO) pigs, inactivating the virus receptor CD163 and pAPN proteins. This method not only prevented PRRSV and TGEV virus infections but also preserved reproductive performance and meat quality [63]. Breeding disease-resistant, gene-edited pigs can reduce economic losses caused by diseases and promote the healthier and more rapid development of the pig industry.

### 3.2. Application of CRISPR–Cas9 Technology in Chicken Genetic Breeding

Due to the unique structure of fertilized chicken eggs, the application of CRISPR–Cas9 in poultry breeding has been less successful thus far than for mammals [64]. Therefore, the application of CRISPR–Cas9 needs to be accelerated in some areas, such as poultry reproductive development and disease resistance breeding [65].

Primordial germ cells (PGCs) in chickens carry genetic information to offspring, but both donor and recipient PGC develop simultaneously in the recipient testes, leading to the rejection of the donor PGC and a reduced sperm count. Researchers have previously used gene-editing technology to construct sterile hosts to eliminate the role of the endogenous receptor PGC. Ballantyne et al. [66] bred DAZL gene-edited chickens using CRISPR–Cas9; in the experiment, the Caspase9 gene was inserted into the chicken PGC at a fixed point, drugs were added to inhibit the growth of PGC in the gonads, and the effectiveness of the infertile hosts was verified via exogenous PGC with regard to feather color. Later, researchers used CRISPR–Cas9 to knockout DMRT1, a gene that determines gonadal differentiation and gender development in male chicken PGC [67]. The edited chicken PGC was injected into the aforementioned sterile host chicken, and the obtained embryos and chicks' gonads developed into ovaries, proving the DMRT1 gene's important role in testicular development [68]. Zhang et al. [69] used CRISPR–Cas9 to knock out the important gene Stra8 (stimulated via recurrent acid gene8) affecting reproductive development in chicken embryonic stem cells, finding that ECS cannot be transformed into male germ cells. MSTN (myostatin) negatively regulates the proliferation and differentiation of skeletal muscle cells [70]. Xu et al. [71] injected the MSTN–knockout sgRNAs' adenovirus vectors into chicks' leg muscles, and MSTN expression was significantly downregulated, though gene-edited offspring were not bred. Therefore, we can conclude that studying the combination of PGC with CRISPR–Cas9 for poultry gene editing has more potential to lead to a scientific breakthrough.

Poultry is highly susceptible to viruses, such as MDV (Mark's disease virus), the highly pathogenic ALV-J (Avian leukosis virus J) [72], and AIV (avian influenza virus) [73], which all have fast transmission speeds and high mortality rates. Koslova et al. successfully bred chNHE1-KO homozygous mutant chickens resistant to ALV infection using CRISPR–Cas9 [74]. Another study used CRISPR–Cas9 to modify the residues of the chANP32A gene to reduce AIV replication [75]. Hellrich et al. [76] used the CRISPR–Cas9 system to knock out Trp38 encoded by chNHE1 (Chicken Na+/H+ exchange type 1) of the ALV-J receptor in chicken PGC, breeding anti-ALV-J chickens as a result. Liu et al. [73] successfully developed a CRISPR–Cas9 vector that blocked the expression of ALV-J in vivo using coinfected MDV, the first time that this had been achieved. In summary, CRISPR–Cas9 gene-editing technology has significantly innovated poultry disease prevention and breeding.

### 3.3. Application of CRISPR–Cas9 Technology in Cattle Genetic Breeding

Cattle are the primary species from which meat, milk, and derivative leather are produced. At present, research into gene editing in cattle mainly covers disease resistance breeding, increasing meat production, and eliminating allergens. By using Cas9 to precisely

insert beneficial genes into animals, researchers can promote the industry's healthy and efficient development, reduce economic losses, and ensure high food safety standards.

Bovine tuberculosis is a zoonosis induced by a *Mycobacterium bovis* (*Mbv*) infection. Gao et al. [77], for the first time, used Cas9 nickase (Cas9 n) to prepare naturally resistant associated macrophage protein-1 (NRAMP1) gene-edited cows, enhancing their resistance to Mycobacterium infection and reducing the risk of human infection through contact with diseased cows or their by-products. Subsequently, Yuan et al. [78] used CRISPR–Cas9 to integrate the NRAMP1 gene into the FSCN1-ACTB (F-A) and bovine homologous mouse Rosa26 sites, enhancing cows' resistance to tuberculosis. Szillat et al. [79] developed a CD46 gene knockout cell line dependent on bovine viral diarrhea virus (BVDV) invasion using CRISPR–Cas9, allowing researchers to determine the mechanism behind CD46's role in plague viruses' replication cycles.

The prevention and treatment of mastitis is a major problem in the dairy industry. Bovine paratuberculosis, caused by Mycobacterium bovis paratuberculosis (MAP), greatly impacts the dairy industry. Using CRISPR–Cas9 to knock out the Interleukin-10 Receptor Alpha (IL-10RA) in bovine mammary epithelial cells that immunoregulate MAP can either inhibit or promote the expression of certain pro-inflammatory cytokines, creating a cellular model that provides a basis for further research into the anti-inflammatory mechanisms of IL-10RA [80].

When horns are present, cattle are prone to accidental injury during play and struggle, which can cause miscarriages in severe cases and even endanger human safety. Therefore, using gene-editing technology to cultivate hornless cattle is a useful approach. Gu Mingjuan et al. [81] used CRISPR–Cas9 to insert the 202 bp repetitive Pc (polling of cellular origin) site of chromosome 1, a gene that controls hornless traits, into the genomes of fibroblasts derived from horned Mongolian cattle, providing basic material for breeding hornless cattle.

The OCT4 gene plays an important role in maintaining the pluripotency of mammals' early embryonic stem cells. Simmet et al. [82] used CRISPR–Cas9 to knockout OCT4 in bovine embryos, finding that its blastocyst development process was similar to that of human blastocysts lacking OCT4. Daigneault et al. [83] targeted the injection of OCT4 into fertilized bovine eggs and found that on day 5, embryonic development stagnated and blastocyst formation was impossible. Based on this, Camargo et al. [84] utilized optimized electroporation technology to transfect CRISPR–Cas9 targeting the sgRNA-Cas9 protein complex of OCT4 into fertilized bovine eggs, reducing operational difficulties and making breeding transgenic animals easier. Their results not only provide an animal model for treating early human embryonic developmental defects but also enable further research into promoting cattle reproduction.

## 4. Application of CRISPR–Cas9 Technology in Goat and Sheep Genetic Breeding

Due to the superior quality of cashmere wool, meat, milk, and other by-products, sheep and goats are also vital livestock in agricultural production and play an important role in animal husbandry. For goats and sheep, gene editing is used to improve growth status, increase cashmere yield and fertility, and breed individuals with multiple excellent traits. Scientists use gene-editing technology to modify different species' traits according to their specific needs to accelerate the process of breeding sheep and goats.

### 4.1. Promotion of Hair Follicle Growth and Development

Cashmere goats are the main local economic primary livestock reared in Inner Mongolia, and cashmere is an important natural raw material for the textile industry. Due to the high economic value of cashmere goats, they are also an important source of income for farmers and herdsmen. Using CRISPR–Cas9 to study hair follicle development-related functions in genes is of great significance for improving the production performance of cashmere-producing goat species.

Studies have shown that using CRISPR–Cas9 to inhibit FGF5 expression can improve cashmere growth in sheep [85,86] and goats [87], increasing hair follicle density and length

and improving quality. Fibroblast growth factor 5 (FGF5) is the main inhibitor that controls fiber length and growth. Wang et al. [87] targeted the knockout of the FGF5 gene in Shaanbei white cashmere goats using CRISPR–Cas9, finding that the number and density of secondary hair follicles in gene-edited cashmere goats significantly increased, as did cashmere length. They also performed HE staining of the skin tissues of aborted and normal goats, showing that the number of secondary hair follicles in the skin of FGF5 mutant goats was higher than that of wild-type goats, the diameters of the primary and secondary hair follicles were longer than those of wild-type goats, and the SHF/PHF ratio was significantly higher than that of the normal goats. Zhang et al. [88] also confirmed that the knockout of FGF5 significantly increased the fiber length and growth rate. Li et al. [89] used CRISPR–Cas9 to insert nonsense codons into the FGF5 gene, which improved the cashmere yield. Hao et al. [90] inhibited hair follicle growth and development in goats by producing SCNT-mediated extracellular receptor abnormalities (EDAR) knockouts using CRISPR–Cas9. In addition, Li [91] added exogenous thymosin β 4 (Tβ4) targeted integration into cashmere goats, discovering that Tβ4 can induce hair follicle development, accelerate the differentiation of hair follicle stem cells, and improve hair production. Hao [90] combined CRISPR–Cas9 and somatic cell nuclear transfer technology to breed EDAR gene-targeted cashmere goats, which exhibited baldness. Hu [86] utilized CRISPR–Cas9 to target vascular endothelial growth factor (VEGF) in Inner Mongolia-bred white cashmere goats with FGF5 and CCR5 genes, which simultaneously knocked in hair follicle development-promoting genes and hair follicle development-inhibiting genes. Wool color is also a key production trait for cashmere. To change the color of cashmere, Zhang et al. [92] applied CRISPR–Cas9 to target and knockout the ASIP gene. Compared with wild-type individuals with white hair, the ASIP-edited sheep wool color produced diversity, indicating that ASIP affects fur color formation and serves as a marker gene. CRISPR–Cas9 is currently able to disrupt genes that inhibit hair follicle development into an ideal phenotype, providing a scientific basis for the further breeding of high-quality and high-yield cashmere goat breeds, as well as providing insights into the treatment of androgenic alopecia.

### 4.2. Improvement of Muscle Growth and Development

Researchers [93–95] used Cas9/gRNA to block genes that inhibit muscle production in meat-producing sheep [96], improving meat production and quality, thus meeting consumer demand for mutton. Myostatin (MSTN) can regulate skeletal muscle growth and promote muscle proliferation. Therefore, inhibiting MSTN expression can induce "dual muscle" traits in developed muscle areas, such as the buttocks, shoulders, and legs, providing application value for improving meat quality and increasing the lean meat percentage in livestock. Albertio and Wolf [97] propose using DNA nucleases to apply this phenotype to cattle, sheep, and pigs, promoting meat production performance in livestock and poultry.

He et al. [98] injected targeted MSTN into the cytoplasm of prokaryotic embryos to breed gene-edited goats with dual muscle traits. Wang et al. [93] demonstrated that the muscle development of MSTN-edited sheep was significantly faster than that of wild-type sheep. Niu et al. [99] found that sheep modified with BCO2 biallelic genes had yellow fat, indicating that the BCO2 gene may determine fat color formation. Wan et al. [100] used CRISPR–Cas9 to knock out the conserved site of the goat CLPG1 gene to study the "beautiful buttocks" of sheep. In addition, Zhang et al. [94] demonstrated that the loss of MSTN function in sheep skeletal muscle satellite cells (sSMSCs) promotes the differentiation and growth of sSMSCs. Zhou et al. [101] introduced single-nucleotide point mutations in the Recombinant Suppressors Of Cytokine Signaling 2 (SOCS2) using CRISPR–Cas9, resulting in an increase in sheep weight and milk production; in 2022, they knocked out the MSTN bialleles with a high proportion of mutations, with meat quality being almost unaffected [102]. For MSTN mutant sheep, not only were changes in transcriptome gene expression analyzed in MSTN knockout goats [103], but we also confirmed how knockout alleles are inherited by offspring [93]. Mei bred one double-allele and one single-allele MSTN gene-mutated goat through the microinjection of fertilized

eggs. Both edited sheep fetuses had "double muscle" and significantly increased weight and length. Further sectioning revealed significant increases in the muscle cross-section and muscle fiber density [104]. CRISPR–Cas9 significantly highlights the functions of genes related to muscle growth and development in sheep, and it has potential roles in the breeding of new breeds of meat sheep.

### 4.3. Improvement of Reproductive Capacity

Improving reproductive performance is a key goal of breeding. The Fec family of genes is associated with the fecundity of sheep. Zhou et al. [105] used CRISPR–Cas9 to inject single-strand FecB DNA oligonucleotides into the zygote of sheep, breeding FecB homozygous mutant sheep, proving that FecB homozygous ewes have not only higher lambing rates but also improved reproductive rates. CRISPR–Cas9 simultaneously edits the HYAL2 and PrP genes to improve the lambing rate [8]. Niu et al. [28] introduced point mutations in goat growth differentiation factor 9 (GDF9) and found that they also affect ovulation rate and litter size. Zhang et al. [106] used CRISPR–Cas9 to induce functional deletion mutations in the bone morphogenetic protein receptor type IB (BMPR-IB) gene in sheep, increasing the ovulation rate and litter size. Based on ssODNs, Zhou et al. [105] further introduced point mutations to breed BMPR-1B gene-edited sheep. Tian et al. [107] microinjected the AANAT gene into frozen and unfrozen sheep embryos, and the results showed that there was no significant difference in the reproductive abilities of transgenic offspring between the two environments, but AANAT transgenic individuals had good reproductive abilities. Thus, CRISPR–Cas9 is an effective editing tool for developing ideal traits in farm animals, promoting the development of animal husbandry.

### 4.4. Improvement of Milk Composition

At present, researchers have enriched the composition of sheep milk by genetically modifying essential nutrients and knocking out non-essential proteins. Goat milk is rich in fat and protein, and its composition is similar to that of human milk, but β-Lactoglobulin (BLG) is prone to sensitization. Zhou et al. [108] found that using CRISPR–Cas9 knockout of β-lactoglobulin in goats reduces the protein concentration in BLG, laying the foundation for improving the composition of goat milk. Wei et al. [109] constructed BLG gene knockout cows from fertilized eggs to broaden the market for milk and its by-products. Another study used CRISPR–Cas9 to knock out factors that affect milk traits, such as Stearoyl CoA Desaturase 1 (SCD1) [110] and Acetyl CoA acyltransferase 2 (ACAA2) [111]. In breast cells, Ma et al. [112] used CRISPR–Cas9 to inject Arylkylamine N-acetyltransferase (AANAT) and Acetylserotonin O-methyltransferase (ASMT), which mediate melatonin expression, into the cytoplasm of fertilized sheep eggs and developed an AANAT/ASMT breast bioreactor, resulting in edited sheep with high yields of melatonin milk. CRISPR–Cas9 is vital to the development of non-artificial, low-fat, or specific functional dairy products, promoting the development of dairy products rich in specific nutrients or pharmaceutical ingredients.

### 4.5. Establishment of Animal Disease Resistance Breeding and Human Disease Models

The economic losses caused by livestock and poultry diseases not only limit the development of animal husbandry but also pose a threat to human safety in the event of a zoonotic disease outbreak. CRISPR–Cas9 has been used in disease resistance breeding for several diseases, providing a good therapeutic model for studying pathogenesis and improving animal health. In addition, the body size and anatomical structure of sheep are similar to those of humans, and CRISPR–Cas9 has been used in sheep to provide an effective model for studying human diseases.

Fan et al. [113] targeted knockout PRP in goat fibroblast donor cells to produce SCNT-mediated anti-PRP goats, and using CRISPR–Cas9 to generate NUP155 gene knockout donor cells in goat fibroblasts, a NUP155 gene knockout goat model for studying heart disease was developed through the SCNT program. Menchaca et al. [114] knocked out hyaluronidase 2 (HYAL2) in lambs, leading to lung adenocarcinoma syndrome, and con-

firmed the possibility of using CRISPR–Cas9 for producing antiviral animals; they also discovered, for the first time, that deleting the sheep otoferlin (OOF) gene is an effective model for treating deafness [115]. Li [116] inserted the Tβ4 gene into the CCR5 site in goats, allowing for the establishment of a goat knock-in model. Fan et al. [117] used CRISPR–Cas9 combined with SCNT technology to breed IFNA gene knockout sheep, providing a large-scale animal model for fetal resistance to Zika virus (ZIKV) infection. Subsequently, they pioneered the human cystic fibrosis (CF) sheep model with CRISPR–Cas9-mediated cystic fibrosis transmembrane transmission regulator (CFTR) deficiency using the same technology [60]. The liver and gallbladder disease phenotypes of newborn CFTR$-/-$ sheep were consistent with those of humans. Williams et al. [118] applied CRISPR–Cas9 to point-mutate the non-specific alkaline phosphatase (TNSALP) gene alkaline phosphatase (ALPL), resulting in edited lambs exhibiting human hypophosphatasia (HPP), providing an effective animal model for studying this rare metabolic bone disease. Vilarino et al. [119] constructed pancreas/duodenum homeobox protein 1 (PDX1) fetuses, allowing for gene-edited sheep to be used as host organs for xenotransplantation. In addition, interspecies blastocyst complementary technology combining embryonic gene editing and pluripotent stem cells (PSCs) achieved xenotransplantation of human organs by using large animals as hosts. The production of these disease-related gene knockout sheep demonstrates the possibility and effectiveness of CRISPR–Cas9 for breeding antiviral livestock, as well as its enormous potential for the treatment of human diseases.

### 4.6. Current Problems with Gene Editing Sheep

Although CRISPR–Cas9 has broad prospects in the development of animal husbandry, the low targeting efficiency, poor chimerism, and off-target effects of CRISPR–Cas9 [120] seriously restrict the sheep breeding process. Scientists are constantly optimizing CRISPR–Cas9 to be more efficient, accurate, and safe. CRISPR–Cas9 has developmental defects and low survival rates when editing cells in vitro to obtain gene-edited homozygotes. Direct injection into the fertilized egg results in the formation of chimeras by simultaneously editing multiple cells due to division. Due to the long pre-attachment period of sheep, the blastocyst during implantation develops a filamentous structure. This evolutionary difference at this stage is considered key to achieving interspecies chimerism. The exploration of interspecies chimerism and blastocyst complementarity methods allows for the production of transplantable, specific organs in large animal hosts, such as sheep. Chimera may occur between different species, but it has not yet been confirmed. In addition, the International Society for Stem Cell Research (ISSCR) and some regulatory agencies have stated that research into interspecies chimeras poses certain ethical questions [121]. Therefore, the electroporation of Cas9 protein and sgRNA into embryos to breed gene-edited sheep not only obtained homozygous positive individuals but also improved editing efficiency. Improving the targeting efficiency of gene-edited sheep is related to various factors, such as increasing the embryo injection concentration of CRISPR–Cas9 or improving the conception rate [122]. For example, the first MSTN mutant sheep bred through embryo microinjection had a gene-editing efficiency of only 5.7%. Later, after several years of research, increasing the embryo injection concentration of the CRISPR–Cas9 system greatly improved the mutation efficiency, and the editing efficiency almost reached 100% [7].

Off-target effects cause unpredictable consequences such as base mutations, deletions, rearrangements, and immune responses [123]. Therefore, reducing the off-target rate is vital for improving CRISPR–Cas9. At present, strategies such as optimizing sgRNA length, modifying Cas9 nuclease specificity, and using other Cas variants are mainly used to reduce off-target effects [124]. Wienert et al. [125] developed a method called DISCOVER-Seq for identifying the off-target effects of Cas. Due to its ability to recruit DNA repair factors in cells and individuals, this method is applicable to various sgRNAs and Cas proteins. Donohue et al. [126] proposed a technique called CRISPR hybrid RNA DNA (chRDNA) to reduce off-target rates while maintaining targeted editing ability and improving Cas9 specificity. Although the efficiency of using viral vectors to deliver CRISPR systems is

high [127], the Cas9 protein is relatively large and must be packaged separately before co-injection. Mout et al. [128] developed engineered DNA-free virus-like particles (eVLPs) that could be efficiently edited in various cells and organs, minimizing the off-target rates and risk of DNA integration. Perez et al. [129] developed an algorithm called the CRISPR Specific Correction (CSC) system for correcting specific gRNAs targeting mismatches in non-coding and repetitive regions. The SpCas9 mutant and other gene-editing techniques have overcome the drawbacks of off-target efficiency in CRISPR–Cas9 [130], but they have also partially changed the species gene pool and ecosystem. Therefore, it is necessary to upgrade and improve more accurate and easily controllable CRISPR–Cas9 systems [131].

In addition to optimizing sgRNA length and improving the Cas9 protein variants mentioned above, scientists are also developing some more creative new technologies. For example, the "GG20" technique of ggX20 sgRNAs containing two guanines makes sgRNAs more specific and can significantly reduce off-target effects [132] by using the CRISPR nickel enzyme to modify a nuclease domain of one of the DNA strands to reduce DNA damage and reduce off-target efficiency [133]. The mechanism of prime editing involves binding the primer-binding site (PBS) of the target to the 5′ region of the primer-editing guided RNA (PegRNA), exposing non-complementary strands. Cas9 nickase (nCas9) cleaves the single-stranded DNA complementary to the template RNA, producing primers for reverse-transcription (RT) enzymes connected to nCas9. At this time, the PegRNA serving as the guided RNA recognizes the target DNA to achieve base-to-base editing [134,135]. At present, the application of prime editor has not caused off-target mutations, but the editor's development is still in its early stages and needs further exploration. Anti-CRISPR (Acr) is a natural inhibitor of the CRISPR–Cas system, and the most direct way to inhibit the action of CRISPR is through binding Acr to the CRISPR–Cas complex, reducing its concentration and, thus, inhibiting its binding to DNA, improving targeting specificity [136]. However, Acr is just a new discovery in the vast CRISPR tool, and there are more new components that scientists need to explore and develop for their application in the field of genetics. Bravo et al. [33] recently created a 7D mutant tool named SuperFi-Cas9, with an editing efficiency (6.3 times) for targeted DNA that is approximately five times higher than that of SpCas9 (1.55 times). It effectively distinguishes on- and off-target substrates, reducing the likelihood of target cleavage by 4000 times [137]. Gene-editing technology currently under development has enormous potential in various fields in the future. It is hoped that the development of SuperFi–Cas9 and other technologies can comprehensively and accurately overcome off-target problems in the near future.

## 5. Conclusions and Prospect

Since the emergence of CRISPR–Cas9, it has greatly advanced modern breeding in agriculture and animal husbandry due to its precision, efficiency, simplicity, and economical nature. As a popular method for studying target genes, the development of CRISPR has solved some new challenges in genetic breeding, production performance, animal health, environmental protection, and human health in modern animal husbandry. Due to technological development, molecular and gene-editing breeding that achieve gene recombination through targeted gene transfer have replaced traditional breeding methods such as artificial selection and multi-generation hybridization. CRISPR–Cas9 can accurately integrate, delete, and replace genes across species; insert excellent trait genes according to needs; break reproductive isolation; and achieve mutual acquisition of disease resistance genes from different species [138], thus reducing the time and cost of long-term breeding, accelerating the breeding process, providing new research ideas for breeding high-quality and high-yield disease-resistant livestock varieties, and creating new production models for gene-edited animals. In addition, developing multiple sgRNA vectors while editing multiple genes with the same individual genome is currently the main research direction for resistance breeding [62]. Despite the many benefits of gene editing breeding, some questions remain: how can we ensure the safety of gene-edited species and their by-products? How can we find effective genes that affect the disease resistance of livestock? Have national

regulators introduced policies for the use and promotion of genetically modified animals? Will humans accept the need to consume genetically modified agricultural products? Once these issues have been resolved, CRISPR gene-editing technology will be even more fruitful in animal husbandry breeding.

The primary problem is that it will cause unavoidable environmental damage. The breeding of gene-edited animals changes the genetic material of species, which may lead to biodiversity imbalance and a reduction in species diversity. For example, when applying gene-editing technology, using certain toxic drugs and reagents, as well as large-scale application by large companies, can cause soil, water, and air pollution and environmental damage, indirectly affecting the survival of other species and directly disturbing the balance of ecosystems.

Secondly, and more importantly, ethical issues must be addressed, as editing animals should be conducted while also meeting their health and natural behavioral needs. Although using gene-editing technology to castrate animals, increase disease resistance, and improve their health and welfare is beneficial, while breeding hornless animals avoids them experiencing harm during fighting, serious off-target effects may lead to animal deformities. Due to our limited understanding of some polygenic diseases, new diseases may emerge or death may occur in animals. Moreover, current studies are not aimed at improving animal welfare but at promoting human health, economic benefits, and convenience of life. These studies are all conducted using gene-editing methods that both improve and harm animal welfare.

At present, the conditions for the gene-editing of farm animals in various countries around the world are not mature. Besides a few industrialized countries such as the United States and Japan, regulatory standards for gene-edited animals in most countries are not clear. Although regulatory agencies attempt to restrict the production of genome-edited livestock and poultry, scientists believe that as long as standardized biosafety testing is passed, it can be applied commercially. Governments have established policies such as the monitoring of breeding and consumption mechanisms for gene-edited agricultural and livestock products, gradually allowing genetically modified animals to enter the market [139]. The European Union monitors the manipulation process and personnel involved in the breeding, supply, testing, and determination of the final destination of genetically edited animals, establishing an animal welfare committee to supervise and manage each link. The personnel involved are assessed and evaluated, and public supervision is publicly accepted. It advocates for bold and innovative research into genetically edited animals based on animal health and prosperity considerations, ensuring safety, careful promotion, and strict legal supervision based on the long-term development and needs of the country, practical applications, and genetically modified research. With the continuous improvement in supervision mechanisms and scientific technology, the public's recognition of genetically modified livestock and poultry products will also further increase. CRISPR–Cas9 whole genome scanning technology will not only save time and cost but also avoid the risk of disease transmission, making it an effective tool for accurately screening resistance genes. However, for disease-resistant traits unsuitable for cell death and virus replication screening, it is currently insufficient to explore the function of their target genes [62]. Although existing CRISPR–Cas9 technology simultaneously edits multiple genes, it cannot do so for over twenty or even hundreds of genes. Thus, in the future, continuous exploration and innovation are needed to optimize the CRISPR screening library and conditions and develop more solid and efficient multi-gene editing technology to achieve the simultaneous regulation of multiple important economic traits of livestock and poultry micro-effect genes [140], thereby breeding the most ideally genetically improved species.

In summary, CRISPR–Cas9 technology has greatly innovated the genetic breeding and improvement of livestock, promoting the sustainable development of animal husbandry. With recent improvements in gene-editing technology and the exploration of the structure and function of Cas9, which further enhances the safety and superiority of gene editing, CRISPR technology's impact has undoubtedly been strengthened. To enable

further innovation and improved protection, we will promote further study into gene-editing technology. We believe that gene-edited animal products will eventually become market-oriented and that the development of human society will allow us to reach a more inclusive and sustainable future. Technological innovation is vital, but it should be used properly, strictly adhering to ethical norms, and researchers and regulators must strengthen public awareness and safety regulations to protect the environment, animal welfare, and human health.

**Author Contributions:** Methodology, Z.L. (Zeyu Lu); validation, Z.L. (Zeyu Lu), L.Z., Q.M. and J.L.; formal analysis, Z.L. (Zeyu Lu); investigation, Z.L. (Zeyu Lu), Y.C. and H.W.; writing—original draft preparation, Z.L. (Zeyu Lu); writing—review and editing, Z.L. (Zeyu Lu); supervision, Y.Z. (Yanhong Zhao), J.L., Y.L., Y.Z. (Yanjun Zhang), R.S., R.W., Z.W., Q.L. and Z.L. (Zhihong Liu); project administration, Y.Z. (Yanhong Zhao); funding acquisition, Y.Z. (Yanhong Zhao). All authors have read and agreed to the published version of the manuscript.

**Funding:** This research was funded by the Science and Technology Major Project of Inner Mongolia (2021ZD0012), the National Natural Science Foundation of China (32160772), the Science and Technology Project of Inner Mongolia Autonomous Region (2023YFHH0076), and the Program for Innovative Research Team in Universities of Inner Mongolia Autonomous Region (NMGIRT2322).

**Institutional Review Board Statement:** This study does not involve humans or animals.

**Data Availability Statement:** No new data were created or analyzed in this study. Data sharing is not applicable to this article.

**Conflicts of Interest:** The funders had no role in the design of this study.

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
