# Peer review of "Progress in Research and Prospects for Application of Precision Gene-Editing Technology Based on CRISPR–Cas9 in the Genetic Improvement of Sheep and Goats"

_agriculture, doi:10.3390/agriculture14030487_

Round 1

Reviewer 1 Report

Comments and Suggestions for Authors

Dear Authors,

Here are some suggestions point to improve the MS

Section of Introduction 

Verify the accuracy of the development history of CRISPR-Cas9, especially the timeline mentioned. CRISPR-Cas9 was not developed until the early 21st century, not the late 1990s.

The introduction needs more detail on why CRISPR-Cas9 is a significant advance over TALEN and ZFN, beyond cost and simplicity.

Consideration is given to the potential ethical and regulatory challenges associated with gene editing in livestock, which provides a more balanced view of the technology's impact.

Section of "2.1. Overview and Principle of CRISPR/Cas9"

The section said that CRISPR was discovered in the genomes of bacteria and archaea. It might be more accurate to state that CRISPR-Cas9 was adapted from a naturally occurring genome editing system in bacteria. The bacteria used CRISPR to fend off invading pathogens like viruses, rather than it being an inherent characteristic of their genomes​​.

The paper states that "CRISPR is composed of Cas9 protein functional genes short regularly clustered repeat sequences and similarly long spacer sequences."  It would be more accurate to say that the CRISPR system includes the Cas9 protein and CRISPR RNA (crRNA), which guides Cas9 to a specific location in the DNA​​.

The section mentions two classes and six types of CRISPR systems. While this is accurate, the paper could be a clearer explanation of the differences between these types and their specific applications, especially since the focus is on the CRISPR/Cas9 system​​.

The authors correctly describe the base pairing principle of the CRISPR/Cas9 system but could elaborate more on how this system can be programmed to target specific DNA sequences, which is central to its use in gene editing​​.

This section lalso discusses the nonhomologous end joining (NHEJ) and homologous recombination (HR) repair mechanisms. However, it is needed to clarify the distinct roles of these mechanisms in CRISPR/Cas9 editing, such as NHEJ often leading to insertions or deletions (indels) at the target site, and HR being used for precise gene insertions​​.

This section also showed the limitations and off-target effects of CRISPR/Cas9 are crucial. Please, include more recent strategies developed to mitigate these issues, providing a more comprehensive view of the current state of CRISPR technology​​.

Section "2.2. Comparison between CRISPR/Cas9 and Other Gene Editing Techniques"

Section 2.2  was written that ZFN recognizes triple bases while TALEN recognizes single nucleotides. Please double-check: ZFNs recognize specific DNA sequences through engineered zinc finger proteins, each typically binding to three base pairs, whereas TALENs use a series of repeat variable di-residues (RVDs) to recognize individual nucleotides. Clarification on how these technologies recognize DNA sequences can improve accuracy​​.

The section 2.2  mentions that TALEN can target longer sequences and has a low probability of missing targets. While this is generally correct, it would be beneficial to add that the length and flexibility of the DNA binding domain in TALENs can be more easily customized than ZFNs. 

The section also emphasizes the simplicity, low cost, and ability of CRISPR/Cas9 to edit multiple genes simultaneously. These are key points, but please further elaborate on how the simplicity of designing sgRNAs for CRISPR/Cas9 compares to the complexity of engineering protein domains in ZFNs and TALENs​​.

While the section mentions off-target effects of CRISPR/Cas9, it could benefit from a more detailed comparison of off-target rates across ZFNs, TALENs, and CRISPR/Cas9. Each of these technologies has different mechanisms that can lead to off-target effects, and mentioning these can provide a more comprehensive comparison​​.

Section "3.1. Application of CRISPR/Cas9 Technology in Pig Genetic Breeding" 

The authors give an example of the use of CRISPR/Cas9 to edit the non-coding region of insulin-like growth factor 2 (IGF2) in Bama pigs. Please clearly specify the type of modifications (e.g., knockouts, insertions) and how these modifications directly contribute to the observed improvements in meat production​​.

For mitochondrial uncoupling protein 1 (UCP1), it should be clear a explanation of how this gene regulates body temperature and affects body fat production in pigs would strengthen the section. Please, write down the mechanism by which UCP1 influences these traits should be explicitly stated​​.

Section 4.6 "Current Problems of Gene Editing Sheep" 

Authors have supported more detailed examples and evidence to mention various issues like low targeting efficiency, poor chimerism, off-target effects, and developmental defects. However, it would be better to include more detailed examples or case studies. This would provide concrete evidence to support the claims and help readers better understand the practical implications of these challenges.

The section could be expanded to include a discussion on the ethical and ecological implications of gene editing in sheep. This includes considerations about animal welfare, potential impacts on biodiversity, and the ethical implications of modifying the gene pool.

Please clearly express the potential future solutions or areas of research that are currently being developed to address these issues.

Given the technical nature of the content, simplifying complex jargon or providing explanations for technical terms could make the section more accessible to readers who are not specialists in the field.

The "Conclusion and Prospect" section of your draft outlines the impacts of CRISPR/Cas9 technology in agriculture and animal husbandry, addressing both the advancements and challenges faced in the field. However, some points could be improved or expanded upon:

While the section mentions issues and risks related to gene editing, such as safety concerns and public acceptance, it could delve deeper into the ethical implications and long-term ecological impacts of gene editing in livestock. However, I suggest that a more thorough discussion of these aspects would provide a more balanced view.

The mentioned of regulatory standards and government policies towards gene-edited animals is an important aspect.  This could be expanded to include a discussion on the current state of international regulations and how different countries are approaching the regulation of gene-edited animals.

The section acknowledges limitations in current CRISPR technology, particularly in editing multiple genes simultaneously. I would suggest expanding on these limitations and the challenges they pose could offer a more realistic perspective on the state of the technology.

Comments on the Quality of English Language

The English language quality of the reviewed paper appears to be generally good, with a clear and formal academic style suitable for scientific literature. 

Author Response

Please see the attachment.  The revised and highlighted manuscript has been uploaded to the system for your review. Thank you.

Reviewer 2 Report

Comments and Suggestions for Authors

Review Result for Manuscript

Tittle:

The Prospect and Application Research Progress of Precision  Gene Editing Technology Based on CRISPR-Cas9 in Livestock Genetics and Breeding

General Overview:

The manuscript under consideration provides an in-depth look at using CRISPR-Cas9 technology in livestock. The authors present a well-organized and informative paper on various aspects of CRISPR-Cas9, particularly its applications in animal genetics.

Strengths:

Comprehensive Data: The manuscript provides an in-depth examination of CRISPR-Cas9 technology, focusing on its applications in livestock. The detailed coverage of animal-specific aspects increases the manuscript's value for readers interested in this field.

Clear Organization: The paper is well-structured, with a logical flow that aids comprehension of the complex CRISPR-Cas9 concepts. Each section is adequately organized, which contributes to a pleasant reading experience.

Insightful Animal Applications: The authors provide helpful information about the practical applications of CRISPR-Cas9 in livestock. The data presented is informative and pertinent to the scientific community working in this field.

Recommendations:

English Language Proofreading: While the content is commendable, the manuscript contains numerous grammatical and punctuation errors. Attention to these language issues is critical for improving the paper's overall readability and professionalism.

Clarity in Expression: Some sentences may require clarification to convey the intended meaning clearly. This will make the paper more accessible to a broader audience.

Comments on the Quality of English Language

english need to be improved

Author Response

(The authors gave the same response as above.)

Reviewer 3 Report

Comments and Suggestions for Authors

I believe the objective of this paper is to review the current status of gene editing in livestock species and perhaps suggest what needs to be done to advance the field. Unfortunately, the English language use is marginal at its best and the "review" is nothing more than a compilation of single sentences with single references that fail to accurately convey the state of the technology in these species. The vast majority of the cited work has still never made it to the "application" stage primarily due to the very limited success of these methods. Few of the works cited ever produce more than a single or few edited animals that have then been used in production agriculture. The review would have a reader believe that "great" progress has been achieved yet this perspective is only based on "one-off" events that have not survived beyond the "research community." The two technologies that are perhaps closest to commercialization are the PRRS-resistant pigs and M.haemolytica leukotoxin-resistant cattle (which were not included in the review). 

The authors would be much better off selecting the topic they are most interested in, which appears to be sheep- and goat-related modifications, and thoroughly reviewing the success or failures with detailed comparisons between studies as to the impact of the editing on production characteristics. Furthermore, if it is the author's choice to continue to attempt to make this a comprehensive review, a table detailing the citation, gene, method, phenotypic outcome, etc... would be much more informative than what is provided in text form. As indicated previously, this appears to be a compilation of single sentences and references condensed into roughly organized paragraphs.

One key point in the Abstract is that the definition of the CRISPR acronym should be correct if you are going to write a review on CRISPRs.

In summary, this is a fair start at a comprehensive review of the current status of the technology but is not well organized, explanatory, or a realistic interpretation of the success in the field.

Comments on the Quality of English Language

Really poor sentence structure and grammar.

Author Response

(The authors gave the same response as above.)

Reviewer 4 Report

Comments and Suggestions for Authors

ü  Highlight the continuous advancements in gene editing technology and its significant impact on livestock breeding.

ü  Acknowledge the successful production of gene-edited pigs, cattle, sheep, and other improved livestock, emphasizing the transformative nature of these developments.

ü  Define gene editing technology as a genetic tool for precise modifications at the genome level, enabling actions such as knocking in, knocking out, deleting, inhibiting, activating, or replacing specific bases of DNA or RNA sequences.

ü  Emphasize the capability of gene editing technology to modify genes accurately at fixed points without the need for DNA templates.

ü  Discuss the widespread use of the CRISPR/Cas9 system in animal genetic breeding research, highlighting its popularity in recent years.

ü  Acknowledge the limitations of the CRISPR/Cas9 system, specifically its relatively low precise insertion efficiency of foreign genes and the presence of off-target effects.

ü  Emphasize the current inadequacy of the CRISPR/Cas9 system for genome editing in large livestock, using cashmere goats as an example.

ü  Present the paper's focus on reviewing the development status, challenges, and application prospects of CRISPR/Cas9-mediated precision gene editing technology in livestock breeding.

ü  Express the need for addressing challenges to enhance the precision and efficiency of the CRISPR/Cas9 system, especially for large livestock.

ü  Discuss the importance of the paper's insights for livestock gene function analysis, genetic improvement, and breeding with local economic characteristics.

ü  Conclude by emphasizing the theoretical reference provided by the review for the advancement of livestock breeding practices.

ü  Ensure clarity and coherence in presenting the information, making the paper accessible to a broad readership interested in the field of gene editing and livestock breeding.

Comments on the Quality of English Language

Moderate editing is needed for English Grammar.

Author Response

(The authors gave the same response as above.)
